# Risk Assessment Considering the Bioavailability of 3-β-d-Glucosides of Deoxynivalenol and Nivalenol through Food Intake in Korea

**DOI:** 10.3390/toxins15070460

**Published:** 2023-07-18

**Authors:** Sang Yoo Lee, Solyi Cho, So Young Woo, Myungsil Hwang, Hyang Sook Chun

**Affiliations:** 1Food Toxicology Laboratory, School of Food Science and Technology, Chung-Ang University, Anseong 17546, Republic of Korea; dm3822@naver.com (S.Y.L.); ssol2021220448@cau.ac.kr (S.C.); mochalatte9@naver.com (S.Y.W.); 2Department of Food & Nutrition, Gachon University, Incheon 21936, Republic of Korea; mskfdahwang63@daum.net

**Keywords:** type B trichothecene, modified mycotoxin, exposure, glucoside conjugate

## Abstract

Deoxynivalenol and nivalenol are major type B trichothecenes and the most frequently occurring mycotoxins worldwide. Their 3-β-d-glucoside forms have recently become a safety management issue. These glucoside conjugates are converted back to the parent toxins during human digestion, but studies to confirm their bioavailability are lacking. In this study, a risk assessment was performed considering the bioavailability of glucoside conjugates. A literature review was conducted to compile the existing bioavailability studies of glucoside conjugates, and three exposure scenarios considering bioavailability were established. As a result of a risk assessment using deterministic and probabilistic methods, both the deoxynivalenol and nivalenol groups had safe levels of tolerable daily intake percentage (TDI%), not exceeding 100%. The TDI% for the nivalenol group was approximately 2–3 times higher than that for the deoxynivalenol group. Notably, infants showed higher TDI% than adults for both toxin groups. By food processing type, the overall TDI% was highest for raw material, followed by simple-processed and then fermented-processed. Since glucoside conjugates can be converted into parent toxins during the digestion process, a risk assessment considering bioavailability allows the more accurate evaluation of the risk level of glucoside conjugates and can direct their safety management in the future.

## 1. Introduction

Deoxynivalenol (DON) and nivalenol (NIV) are the major forms of type B trichothecene mycotoxins and are among the most frequently occurring mycotoxins worldwide. DON and NIV are principally produced by *Fusarium* species, such as *F. graminearum* and *F. culmorum*. They mainly contaminate grains, such as maize and wheat, and are also found in processed foods, such as bread and beer. DON and NIV are also known as vomitoxin because they cause vomiting when ingested. Chronic exposure may cause immunotoxicity and hepatotoxicity. NIV is reportedly more toxic than DON, yet it only differs in structure from DON by possessing a C-4 hydroxyl (OH) group [1]. In response to fungal contamination and mycotoxin production, several cereal crops have been found to transform mycotoxins into mycotoxin metabolites, such as sugar conjugates, via the plant’s defense mechanism. The co-occurrence of these masked or modified mycotoxins with parent mycotoxins in cereals has been confirmed. Deoxynivalenol-3-β-D-glucoside (DON3G) and nivalenol-3-β-D-glucoside (NIV3G) are the major modified forms of type B trichothecenes [2,3]. Modified mycotoxins may be hydrolyzed and converted back to their parent mycotoxins during digestion, posing a potential additional safety management issue.

Among type B trichothecenes, DON is the major contaminant of cereals worldwide, except in East Asian countries, such as Korea and Japan, where contamination with NIV is similar to or higher than that of DON. Shin et al. [4] confirmed that most genotypes of fungi of the *Fusarium* species isolated from cereals in Korea are of the NIV type. Because there is no commercial standard for NIV3G, contamination by glucoside conjugates is mainly reported for DON3G. The level of contamination by glucoside conjugates varies by food but is reportedly approximately 30% of the level of the parent toxin [5]. Ok et al. [6] analyzed DON and NIV in white (*n* = 241) and brown rice (*n* = 241) in Korea. They detected DON and NIV in the range of 7.1–372 and 12.6–2171.7 μg/kg in 5% and 21% of the white rice samples and at 9.1–434.5 and 17.3–2533.9 μg/kg in 7% and 34% of the brown rice samples, respectively. Gab-Allah et al. [7] analyzed DON, DON3G, and NIV in cereals (*n* = 67), including maize and wheat, and detected levels in the range of 0.48–1223.0, 0.14–419.1, and 0.45–234.1 μg/kg in 94%, 88%, and 86.5% of the samples, respectively, in Korea. Lee et al. [8] analyzed DON, NIV, and their glucoside conjugates in grains, pulses, and their processed products (*n* = 506) in Korea. DON, DON3G, NIV, and NIV3G were detected in the range of 2.0–1018.4, 4.5–93.6, 4.6–370.8, and 7.6–250.6 μg/kg in 13%, 8%, 12%, and 5% of samples, respectively.

The ingestion of food is a major route of human exposure to mycotoxins. To protect consumer health from the risk of type B trichothecene exposure, the United States, the European Union, Japan, and Korea have followed the Codex guidelines for mycotoxin food safety risk by setting a maximum limit of DON contamination of 1000 μg/kg. Meanwhile, no country has yet established and regulated a maximum limit for NIV. The Joint Food and Agriculture Organization/World Health Organization Expert Committee on Food Additives (JECFA) has set a tolerable daily intake (TDI) of 1 μg/kg body weight (bw)/day for the sum of DON and its 3-acetyl- and 15-acetyl derivatives [9], whereas the calculated TDI for NIV differs by institution. The Food Safety Commission of Japan (FSCJ) and the European Food Safety Authority (EFSA) have set a TDI of 0.4 and 1.2 μg/kg bw/day for NIV, respectively [10,11]. In 2017, the EFSA proposed a group TDI for NIV and DON and their 3-β-D-glucoside forms. Due to the lack of studies on the hydrolysis and absorption of glucoside conjugates, the EFSA recommended assuming that 3-β-D-glucosides are metabolized to DON and NIV, absorbed to the same extent as the parent compound, and exhibit similar acute and chronic adverse health effects [2,3]. EFSA conservatively assumed that the relative bioavailability of these glucoside conjugates (modified mycotoxin) was 100%, but the bioavailability of glucoside conjugates may be lower or higher than 100% and may also vary with age. Therefore, the overall risk with parent toxin may change depending on the bioavailability of the glucoside conjugates, and risk assessment considering the bioavailability is important.

Subsequently, Gratz et al. [12] confirmed that DON3G and NIV3G were stable in artificial digestive fluid conditions modeling the mouth, stomach, and small intestine and were largely hydrolyzed by intestinal microorganisms through fecal fermentation. A transwell absorption model using Caco-2 cells also established that the glucoside conjugate was not absorbed. Additionally, Catteuw et al. [13] did not detect DON3G in the hepatic portal vein after the oral administration of DON3G in pigs, and only recovered DON. In summary, glucoside conjugates of DON and NIV are considered stable in digestive fluids but are hydrolyzed by intestinal microorganisms. Because glucoside conjugates are already detoxified by the plant’s defense mechanism, their toxicity is lower than that of the parent toxins. In addition, the glucoside conjugates of DON and NIV are not absorbed intact from the intestine but rather are absorbed only after being converted back to the parent toxin. Therefore, it is important to assess the risk of glucoside conjugates by checking their hydrolysis and absorption in vivo.

Vidal et al. [14] confirmed the absorption and metabolism of DON and DON3G in humans (>18 years, *n* = 20). A meal containing the toxin was provided, and urine samples were monitored for 24 h after the meal. For DON and DON3G, 64.0 ± 22.8% and 58.2 ± 16.0% of the doses were recovered, respectively. The bioavailability of DON3G was not significantly different from that of DON. As mentioned above, Catteuw et al. [13] administered a toxin-containing diet to pigs (4 weeks, infants, *n* = 8). Blood samples were monitored for 48 h after dosing. For DON and DON3G, 77.9 ± 47.7% and 83.0 ± 69.1% of the doses were recovered, respectively. It was confirmed that the absorption of intact DON3G did not occur, and the bioavailability of DON3G did not differ significantly from that of DON. Broekaert et al. [15] monitored blood samples after administering a toxin-containing diet to pigs (11 weeks, adults, *n* = 11). Recoveries of 81.3 ± 17.4% for DON and 16.1 ± 5.4% for DON3G were reported. The bioavailability of DON3G was about 20% compared with DON. Overall, these studies show that the bioavailability of DON3G is similar to that of DON, but with age-related differences. To estimate the health risk arising from the consumption of glucoside conjugates of DON and NIV, a risk assessment considering bioavailability in various, much more realistic scenarios is necessary [16,17].

In general, mycotoxin contamination is left-censored data with many undetected samples. A large number of undetected samples makes it difficult to estimate the distribution for probabilistic risk assessment. Accordingly, most risk assessments have been conducted with deterministic methods, which are generally considered more conservative than probabilistic methods [18,19]. Recently, studies using both deterministic and probabilistic methods have been conducted to estimate the risk of mycotoxins more accurately [20,21].

In this study, three exposure scenarios were set, considering the absorption rate of glucoside conjugates, in order to evaluate the risk of type B trichothecenes, including glucoside conjugates. The risk was characterized using occurrence data of 537 cereals, legumes, and their processed products investigated in previous studies. For systematic risk assessment, exposure was estimated for each type of food processing, and both deterministic and probabilistic approaches (Monte Carlo simulation method) were used.

## 2. Results and Discussion

### 2.1. Estimated Exposure to DON, NIV, and Their Glucosides from Each Food

The deterministic method was used to evaluate the exposure to the DON group (DON and DON3G) and the NIV group (NIV and NIV3G) based on the consumption of each food. For all age groups (scenario 1 and 2), the calculated exposure to the DON and NIV group was 0.0195–0.3159 and 0.0179–0.3317 μg/kg bw/day, respectively. In the case of the DON group, the exposure of 0.02–0.32 μg/kg bw/day calculated in this study was similar to or lower than the 0.2–3.7 μg/kg bw/day evaluated by EFSA and the 0.16–0.74 μg/kg bw/day evaluated by Gab-Allah et al. in Egypt [2,22]. It was also similar to the 0.39 μg/kg bw/day based on human urine biomonitoring data evaluated in Portugal [23]. In the case of the NIV group, the exposure of 0.02-0.33 calculated in this study was similar to the 0.05–0.23 μg/kg bw/day evaluated by Gab-Allah et al. for grain-based food in Egypt [22]. Overall, exposure to the NIV group was about 2–3 times higher than the DON group. In both the DON and NIV groups, the exposure of infants was higher than adults (DON group: 0.0155–0.3252 vs. 0.0211–0.2501 μg/kg bw/day, NIV group: 0.0212–0.3809 vs. 0.0180–0.2463 μg/kg bw/day).

The food items that contributed to the exposure were investigated by age group. In all age groups, the pattern was the same as that of adults and was not expressed separately. Both the DON and NIV groups showed high exposure contributions from cereals and cereal products, such as barley and maize (in all age groups, scenario 1 and 2).

Exposure to the DON group from beer was very high (39%) (Figure 1A), similar to that reported previously [24]. In particular, beer has a high level of contamination of glucoside conjugates. Contaminated DON can be converted to DON3G by metabolic enzymes activated during the germination process of malt, or the yeast used in beer fermentation can convert DON to DON3G [25]. The next-highest contribution to the DON group was barley (12%), followed by maize (11%) and ramen (10%). In infants, maize contributed the most, with 29%, followed by barley (22%), foxtail millet (13%), and sorghum (8%) (Figure 1B). Both adults and infants showed a high exposure to the DON group from cereals and cereal products. Beer presented a higher exposure because of its higher intake in the adult group. Because infants do not consume beer, it did not contribute to the exposure of infants, whereas the contributions from exposure to sorghum and breakfast cereal were higher than those of adults. Unlike the DON group, contamination with the NIV group was confirmed in pastes, such as gochujang and mixed paste. For the NIV group, wheat flour (48%) was the food that contributed the most to adult exposure (Figure 1C), followed by barley (20%) and gochujang (14%). In infants, wheat flour (42%), barley (25%), grain-based baby food (7%), and foxtail millet (6%) were the top four contributors (Figure 1D). For the NIV group, the food items contributing to the exposure of adults and infants were similar.

In the probabilistic method, the distribution of exposure from each food could not be calculated due to insufficient occurrence data to estimate the distribution.

#### Contribution of Glucoside Conjugates to Exposure

The contribution of exposure to glucoside conjugates was investigated in three scenarios. Assuming the average intake and the 95% extreme intake, the overall contribution was similar, so only the contribution under the assumption of the average intake is shown in Figure 2. The contribution of DON3G was 35–51% in scenario 1, 21–35% in scenario 2, 14–22% in scenario 3 (adults), and 9–43% in scenario 3 (infants). The contribution of NIV3G was 32–50% in scenario 1, 19–33% in scenario 2, 11–20% in scenario 3 (adults), and 22–45% in scenario 3 (infants). Both DON3G and NIV3G have been shown to contribute to up to about 50% of the total exposure, and infants are particularly vulnerable to exposure to glucoside conjugates compared with adults. Exposure to glucoside conjugates may be elevated in infants because of the difference in their gut microbiome compared to adults and their immature gut immune system [13,26].

### 2.2. Estimated Health Risk (TDI%) of DON, NIV, and Their Glucosides for Each Food

Based on the estimated exposure, the TDI% of the DON and NIV groups was calculated (Table 1 and Table 2). For the DON group, in scenario 1, the calculated TDI% was 2.3–7.2% (minimum lower bound–maximum upper bound, LB–UB) and 12.2–31.6%, respectively (Table 1). The corresponding TDI% values in scenario 2 were 1.9–5.3% and 10.2–23.8%, respectively. In scenario 3, the calculated TDI% were 2.1–5.1% and 12.4–25.0% for adults and 1.5–8.7% and 8.0–32.5% for infants, assuming the average and the 95% extreme intake, respectively. For the NIV group, in scenario 1, the calculated TDI% was 5.4–18.8% and 28.4–82.9%, assuming the average and the 95% extreme intake, respectively (Table 2). The corresponding TDI% values in scenario 2 were 4.5–14.1% and 23.7–62.6%, respectively. In scenario 3, the calculated TDI% were 4.5–13.2% and 24.0–61.6% for adults and 5.3–25.2% and 26.9–95.2% for infants, assuming the average and the 95% extreme intake, respectively. In all scenarios, the TDI% for the DON and NIV groups did not exceed 100%. Comparing adults (relative bioavailability: 25%) and infants (relative bioavailability: 75%), infants were at a high risk of exposure to both the DON and NIV groups compared with adults, in accordance with a previous study [27]. In particular, the TDI% of the NIV group was calculated as 95.2% (close to 100%) in the scenario assuming extreme food intake in the infant group. The higher risk of infants was due to the higher exposure per unit bw due to lower bw. Overall, the risk of exposure to the NIV group was about 2–3 times higher than that of the DON group. This seems to reflect the high NIV contamination in foods from Korea and the high toxicity of NIV (lower TDI than DON) [1,6,28,29].

### 2.3. Estimated Health Risk (TDI%) of DON, NIV, and Their Glucosides by Food Processing Type

Deterministic and probabilistic methods were applied to assess the risk of exposure to the DON and NIV groups according to the food processing type. From the estimation of the risk to the DON group using the deterministic method, TDI% ranges of 0.8–13.1% (mean intake with LB data–95% intake with UB data), 0.4–8.9%, 1.2–9.6%, and 2.3–31.6% in scenario 1, and 0.7–10.3%, 0.4–7.0%, 0.8–6.6%, and 1.9–23.8%, in scenario 2 were calculated for the raw sample, the simple processed sample, the fermented sample, and the total sample, respectively (Figure 3). The corresponding values in scenario 3 were 0.7–9.1%, 0.4–7.0%, 0.9–8.9%, and 2.1–25.0% for adults, and 1.1–20.0%, 0.3–11.4%, 0.1–1.1%, and 1.5–32.5%, for infants, respectively. For the NIV group, the TDI% ranges of 1.6–35.0%, 2.8–33.9%, 1.0–14.0%, and 5.4–82.9% in scenario 1 and 1.5–26.4%, 2.2–25.4%, 0.8–10.9%, and 4.5–62.6% in scenario 2 were calculated for the raw sample, the simple processed sample, the fermented sample, and the total sample, respectively. The corresponding values in scenario 3 were 1.4–22.7%, 2.2–24.6%, 0.9–14.3%, and 4.5–61.6% for adults, and 2.0–49.9%, 3.0–41.9%, 0.3–3.6%, and 5.3–95.2% for infants, respectively. In all scenarios, none exceeded 100% of the TDI.

The fit distribution of each parameter for the simulation of the probabilistic method is presented in Appendix A. In most cases, the log-normal distribution was found to be the best fit. From the estimation of the risk to the DON group using the probabilistic method, the average TDI% was calculated to be 26.3%, 3.7%, 2.3%, and 18.5% in scenario 1 and 23.3%, 2.7%, 1.8%, and 14.7% in scenario 2 for the raw sample, the simple processed sample, the fermented sample, and the total sample, respectively (Figure 4). The corresponding values in scenario 3 were 29.1%, 2.8%, 3.2%, and 16.2% for adults and 46.2%, 5.9%, 0.6%, and 24.8% for infants, respectively. The estimated average exposure did not exceed 100% of the TDI. However, in the infant group, the 95% extreme value of the raw sample group was calculated to be 159%, thus exceeding 100% of the TDI. For the NIV group, the TDI% was calculated to be 54.9%, 9.9%, 7.1%, and 52.4% in scenario 1 and 27.7%, 6.8%, 5.1%, and 39.2% in scenario 2 for the raw sample, the simple processed sample, the fermented sample, and the total sample, respectively. The corresponding values in scenario 3 were 65.6%, 7.0%, 9.2%, and 42.2% for adults and 60.3%, 15.8%, 1.6%, and 65.0% for infants, respectively. The estimated average exposure did not exceed 100% of the TDI. However, in the case of the 95% percentile value, the raw sample (188%) and total sample (168%) in scenario 1, the total sample (128%) in scenario 2, the raw sample (223%) and total sample (140%) in scenario 3 (adults), and the raw sample (152%) and total sample (209%) in scenario 3 (infants) all exceeded 100% of the TDI.

Overall, for both the DON and NIV groups, the raw sample showed the highest TDI%, followed by the simply processed sample and the fermented sample. This result can be explained by previous research, showing that DON and NIV can be physically and chemically removed during food processing [30,31]. Both the deterministic and probabilistic methods showed higher levels of risk for infants than adults, likely due to the lower bw in the infant group. In the case of deterministic risk assessment, statistical analysis could not be performed because there was no variance in the data due to the point estimation method. In the case of the probabilistic method, the TDI% of the DON group through the total food intake of the adults and infants were 16.2% ± 152.2% and 24.8% ± 153.1%, respectively, showing statistically significant differences (*p* < 0.05). The TDI% of the NIV group also showed a significant difference (*p* < 0.05) between the adults (42.2% ± 379.1%) and infants (65.0% ± 356.3%). Deterministic methods are generally more conservative than probabilistic methods [18,19]. However, the exposure level estimated with the probabilistic method in this study was higher than that estimated with the deterministic method. In the deterministic method, the mean and 95% extreme values were used for the intake, but only the mean value was used for the contamination level. Thus, it is assumed that a lower exposure was estimated with the deterministic method than the probabilistic one.

A sensitivity test was conducted to confirm the contribution of each factor to the risk calculated with the probabilistic method [32]. As a result, for all age groups (scenario 1 and 2) and adults (scenario 3 [adults]), food intake contributed the most to risk (Figure 5). For infants (scenario 3 [infants]), the risk contribution levels of occurrence and food intake were similar. Therefore, more thorough control over the mycotoxin contamination of foods consumed by infants is necessary.

The probabilistic method estimates the distribution of the population based on the distribution of the sample group. It is difficult to accurately estimate the population distribution unless the number of sample groups exceeds an appropriate level. Mycotoxin occurrence is left-censored data with many zero occurrence samples (<limit of detection, LOD), generally. Therefore, if the number of analyses performed for one food is large and the number of positive samples is insufficient, then applying a probabilistic method may not be appropriate. Using deterministic and probabilistic methods in parallel can be a useful approach to accurately assess and manage the risk of mycotoxins through food intake.

## 3. Conclusions

The purpose of this study was to conduct a risk assessment considering the bioavailability of type B trichothecene glucoside conjugates. Among the 537 cereals, legumes, and their products investigated in previous studies, DON, DON3G, NIV, and NIV3G occurrence data were used. The bioavailability of the glucoside conjugates was confirmed with a literature review, and three exposure scenarios were established. From a risk assessment using deterministic and probabilistic methods, the TDI% of both the DON and NIV groups was at a safe level, not exceeding 100%. The TDI% of the NIV group was about 2–3 times higher than that of the DON group. Since infants showed a higher TDI% than adults for both toxin groups, more caution is required. The TDI% by food processing type tended to decrease in the order of raw material > simply processed > fermented processed. The risk assessment considering the bioavailability of DON3G and NIV3G performed in this study was the first attempt to accurately evaluate the risk of glucoside conjugates and suggest a direction for the safety management of glucoside conjugates. Nevertheless, a limitation of this risk assessment was that it is based on insufficient previous studies to definitively determine the bioavailability of glucoside conjugates. Another limitation is that the risk was assessed based on data on the occurrence of a small number of glucoside conjugates. In the future, if accurate bioavailability is confirmed through the accumulation of in vivo studies on the absorption of glucoside conjugates, more accurate risk estimation will be possible.

## 4. Materials and Methods

### 4.1. Hazard Identification

To calculate the TDI for DON, the EFSA derived a benchmark dose for a 5% response (BMDL_05_) of 0.11 mg/kg bw/day after using the weight gain/loss symptoms reported in a chronic toxicity study conducted for 2 years in mice [33]. The TDI of 1.0 μg/kg bw/day was established considering the uncertainty coefficient of 100 for interspecies and intraspecies variability in the calculated BMDL_05_ value [2].

The TDI for NIV was set as 0.4 μg/kg bw/day by the FSCJ and 1.2 μg/kg bw/day by the EFSA [3,10]. Therefore, in this study, the TDI for NIV was newly established by combining reports from safety management organizations such as the FSCJ, EFSA, and the United States Environmental Protection Agency (US EPA), as well as recent studies. Dose–response data from experimental animal studies were analyzed with the no-observed-adverse-effect-level approach and the statistically significant approach with benchmark dose (BMD) analysis using PROAST software (version 38.9, RIVM, Bilthoven, The Netherlands) developed by the National Institute for Public Health and the Environment in the Netherlands. In animal toxicology studies, the critical effects were immunotoxicity and hemotoxicity, which were proven in several species. Three rat and four mouse studies were selected to determine a point of departure (POD) for NIV (Appendix A) [34,35,36,37,38,39,40]. Takahashi et al. [37], Kubosaki et al. [38], and Sugita-Konishi et al. [39] all showed a dose-dependent decrease in white blood cell (WBC) counts. Those same groups conducted repeat-dose toxicity studies in rats, and the lowest-observed-adverse-effect level of 0.4 mg/kg bw/day was determined. In the BMD analysis, 0.36 mg/kg bw/day was determined to be the lowest BMDL_05_ from the data, which showed a decreasing number of WBCs in males and females. The BMDL_05_ of 0.36 mg/kg bw/day for a POD was selected (Appendix A). Deriving the health-based guidance value, an uncertainty factor (UF) of 10 for interspecies differences and a UF of 10 for intraspecies differences were used. The additional factor of 10 for uncertainty in the database was considered.

Therefore, an overall UF of 1000 to BMDL_05_ 0.36 mg/kg bw/day and a TDI for external oral exposure to NIV in humans of 0.4 μg/kg bw/day were established based on the reduction in WBCs in rats. The following TDI was obtained:(1)$$TDI\;for\;NIV=\frac{{BMDL}_{05}0.36\;mg/kg\;bw/day}{UF\;1000}=0.4\;\mu g/kg\;bw/day$$

### 4.2. Exposure Analysis

#### 4.2.1. Occurrence Data

Occurrence data for risk assessment were analyzed using the results of Lee et al. [8]. A total of 537 cereals, legumes, and their processed products collected in 2017–2018 from the Korean market (the four cities of Anseong, Anyang, Seoul, and Uiwang) were analyzed.

#### 4.2.2. Determination of Scenarios Considering Absorption Rate of Glucoside Conjugate

To set up a scenario considering the absorption rate of glucoside conjugates, previous studies were investigated using the keywords “deoxynivalenol”, “nivalenol”, “absorption”, and “glucoside”. After reviewing titles and abstracts, seven studies were identified that could confirm the relative bioavailability of glucoside conjugates (DON3G and NIV3G) compared with their parent toxins (Table 3). Human data were considered first, followed by animal data. Among animal data, pigs were given priority, based on their similar metabolic systems and susceptibility compared to humans, and models with different metabolic systems or susceptibility were not reflected in the scenario setting. According to the priorities, scenario 1 was set as 100% relative bioavailability of glucoside conjugates by referring to Vidal et al. [14] as human studies. Scenario 2 was set with a relative bioavailability of 50% of glucoside conjugates by referring to Catteuw et al. [13], Broekaert et al. [15], and Nagl et al. [41] as animal (pig) data. In scenario 3, the relative bioavailability of 25% for adults and 75% for infants was set considering the difference in bioavailability according to age in the animal (pig) data.

### 4.3. Risk Characterization

#### 4.3.1. Deterministic Approach

The deterministic method was used to estimate the risk of type B trichothecene exposure through food consumption. The risk was characterized by calculating the estimated daily intake (EDI) and comparing it with the established TDI. The formula for calculating the EDI is as follows: EDI (μg/kg bw/day) = occurrence (μg/kg) × daily food consumption (g/day)/bw (kg). Mycotoxin occurrence is generally left-censored data with many undetected samples. To utilize left-censored data, LB (<LOD = 0) and UB (<LOD = LOD) approaches were used [44]. Food intake and bw refer to the Seventh Korea National Health and Nutrition Examination Survey (KNHANES) data. To determine the risk of the DON group and the NIV group, the TDI% was calculated using the following formula: EDI (μg/kg bw/day)/TDI (μg/kg bw/day) × 100. The TDI of 1.0 μg/kg bw/day for the DON group was used, as set by the JECFA, and the TDI of 0.4 μg/kg bw/day for the NIV group was calculated in this study.

#### 4.3.2. Probabilistic Approach

The probabilistic method was used to estimate the distribution of exposure to type B trichothecenes through food intake. Monte Carlo simulation was used to estimate the distribution. Each food was classified into three groups according to the type of processing: raw material, simply processed (e.g., ground, baked, and heated), and fermented processed. In the case of LB data, there were many undetected samples, so it was difficult to secure minimum data for distribution estimation. Therefore, the distribution was estimated based on UB data. Crystal ball software (version 11.1.1.1, Oracle, Austin, TX, USA) was used for the simulation, and the distribution was estimated through 100,000 trials by selecting a distribution model with a high fit. Parameters used in the simulation model included occurrence, consumption, and bw. To confirm the contribution of each parameter to the estimated human health risk, a sensitivity test using the rank correlation coefficients of each parameter was performed.

### 4.4. Statistical Analysis

Statistical analysis was used to compare TDI% in adults and infants. In the case of the deterministic method, statistical analysis was not applied because it was the point estimation method. A Student’s *t*-test was performed using the mean and standard deviation of TDI% for adults and infants, calculated using a probabilistic method. If the *p*-value was less than 0.05, the two groups were judged to have a statistically significant difference.

## Figures and Tables

**Figure 1 toxins-15-00460-f001:**
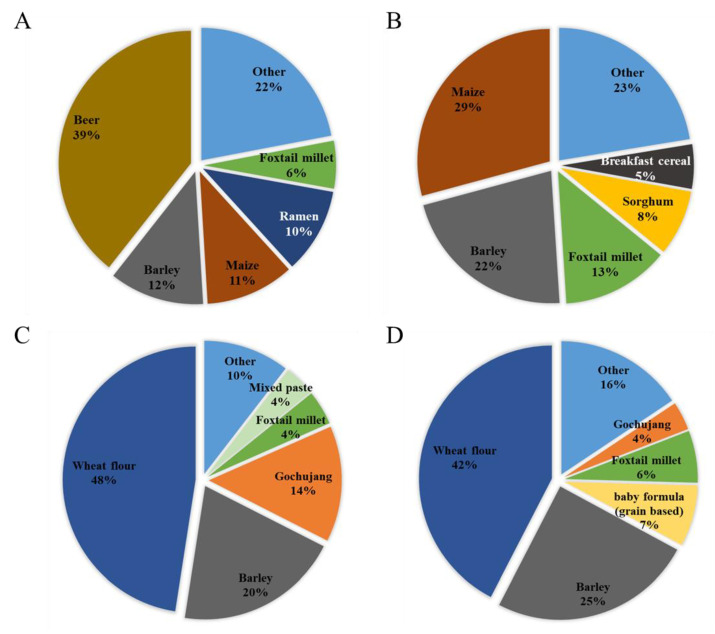
Contribution of each food to mycotoxin exposure in adults and infants (scenario 3). (**A**) DON group (adults); (**B**) DON group (infants); (**C**) NIV group (adults); and (**D**): NIV group (infants). DON, deoxynivalenol; NIV, nivalenol.

**Figure 2 toxins-15-00460-f002:**
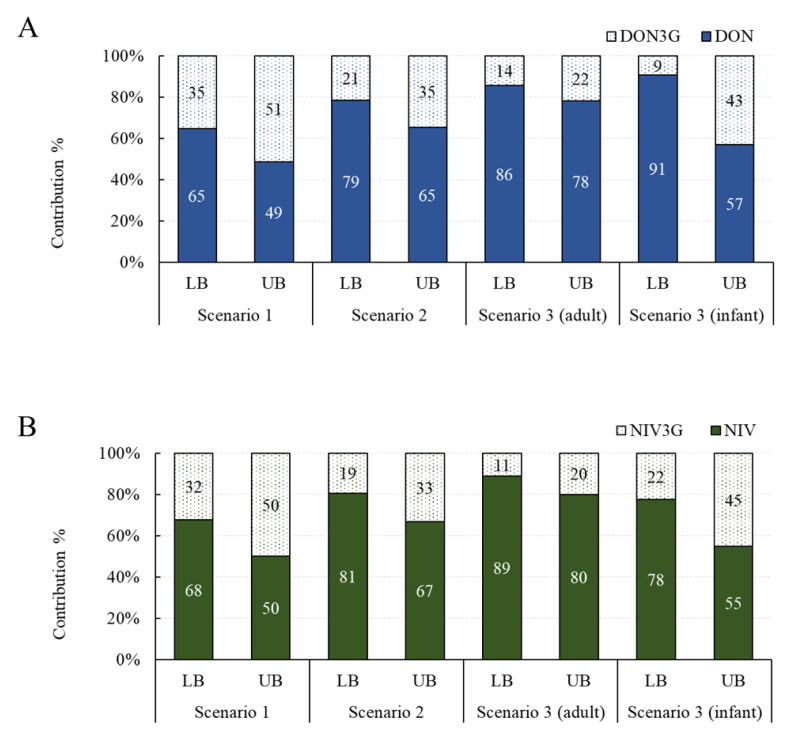
Exposure contribution of glucoside conjugates by scenario. (**A**) DON group; (**B**) NIV group. DON, deoxynivalenol; NIV, nivalenol.

**Figure 3 toxins-15-00460-f003:**
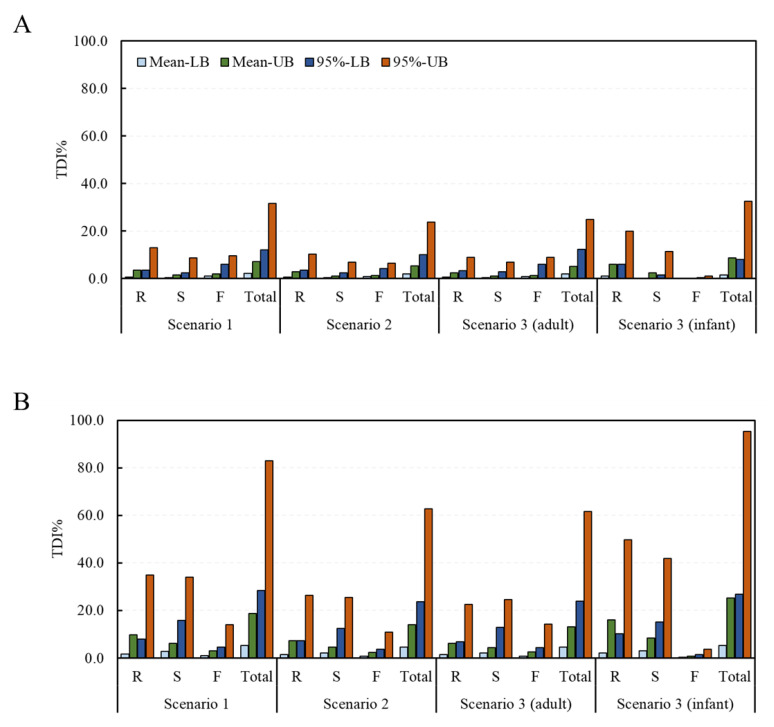
TDI% of the DON and NIV groups estimated by food processing type (deterministic approach). (**A**) DON group; (**B**) NIV group (mean: average food intake, 95%: 95% percentile extreme food intake, LB: LB occurrence, UB: UB occurrence). Raw sample (R), simple processed sample (S), fermented sample (F), and total sample (Total). TDI, tolerable daily intake; DON, deoxynivalenol; NIV, nivalenol; LB, minimum lower bound; UB, maximum upper bound.

**Figure 4 toxins-15-00460-f004:**
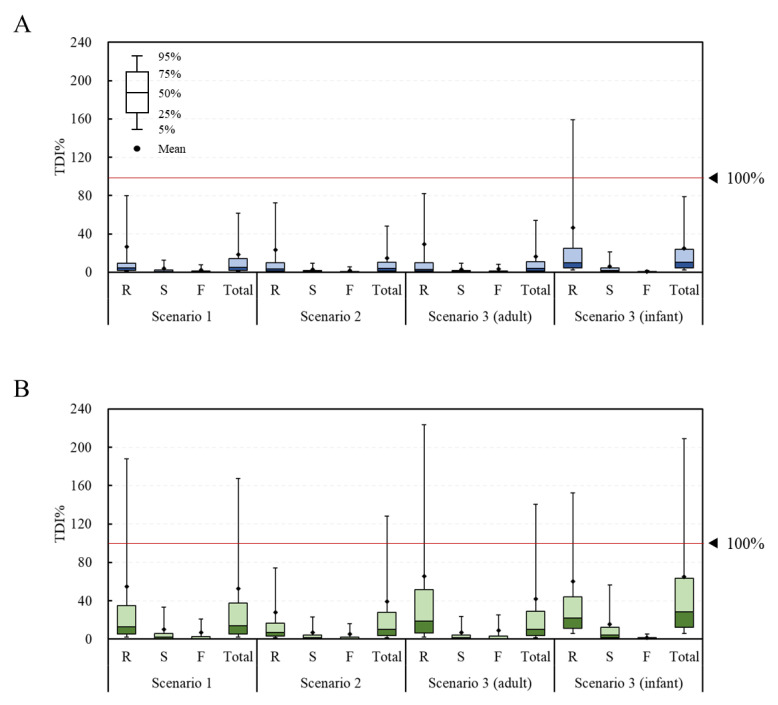
TDI% of the DON and NIV groups estimated by food processing type (probabilistic approach). (**A**) DON group; (**B**) NIV group. Raw sample (R), simple processed sample (S), fermented sample (F), and total sample (Total). TDI, tolerable daily intake; DON, deoxynivalenol; NIV, nivalenol.

**Figure 5 toxins-15-00460-f005:**
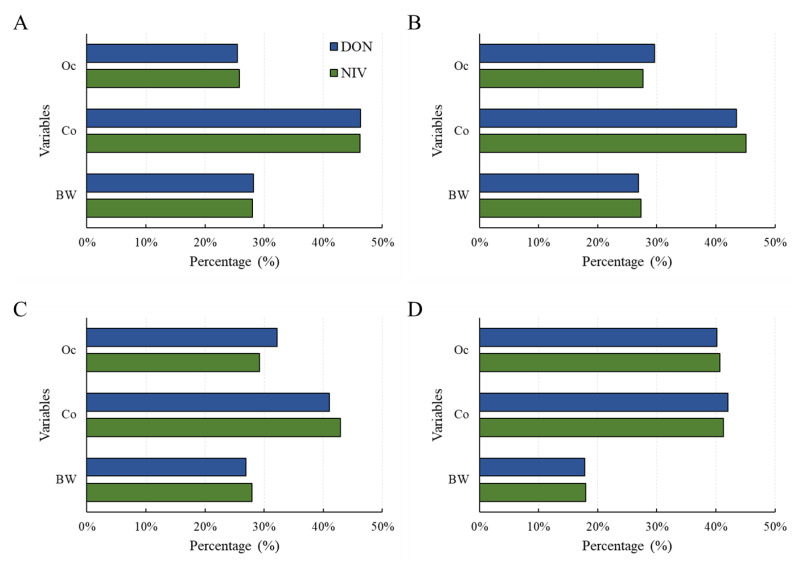
Sensitivity analysis for health risk of the DON and NIV groups according to the three scenarios: (**A**) scenario 1; (**B**) scenario 2; (**C**) scenario 3 (adults); (**D**) scenario 3 (infants). Food intake (Co), Occurrence (Oc), body weight (bw). DON, deoxynivalenol; NIV, nivalenol.

**Table 1 toxins-15-00460-t001:** Estimated TDI% of DON group by three scenarios according to glucoside bioavailability.

Food	Scenario 1				Scenario 2				Scenario 3 (Adult)				Scenario 3 (Infant)			
	Mean ^1^		95% ^2^		Mean		95%		Mean		95%		Mean		95%	
	LB ^3^	UB ^4^	LB	UB	LB	UB	LB	UB	LB	UB	LB	UB	LB	UB	LB	UB
Baby formula ^5^	0.0	0.0	0.0	0.0	0.0	0.0	0.0	0.0	0.0	0.0	0.0	0.0	0.0	0.4	0.2	2.3
Baby formula ^6^	0.0	0.0	0.0	0.0	0.0	0.0	0.0	0.0	0.0	0.0	0.0	0.0	0.0	0.0	0.0	0.0
Barley	0.3	0.3	1.4	1.8	0.3	0.3	1.3	1.6	0.2	0.3	1.3	1.5	0.3	0.4	1.6	2.0
Barley tea	0.0	0.1	0.0	0.2	0.0	0.0	0.0	0.1	0.0	0.0	0.0	0.1	0.0	0.1	0.0	0.1
Beer	1.1	1.4	5.7	7.6	0.7	1.0	3.8	5.1	0.8	1.1	5.6	7.4	0.0	0.0	0.0	0.0
Breakfast cereal	0.0	0.0	0.1	0.1	0.0	0.0	0.1	0.1	0.0	0.0	0.0	0.1	0.1	0.2	0.2	0.4
Brown rice	0.0	0.1	0.0	0.7	0.0	0.1	0.0	0.5	0.0	0.1	0.0	0.4	0.0	0.1	0.0	0.8
Buckwheat	0.0	0.0	0.0	0.0	0.0	0.0	0.0	0.0	0.0	0.0	0.0	0.0	0.0	0.0	0.0	0.0
Cheonggukjang	0.0	0.0	0.0	0.0	0.0	0.0	0.0	0.0	0.0	0.0	0.0	0.0	0.0	0.0	0.0	0.0
Chunjang	0.0	0.0	0.1	0.1	0.0	0.0	0.1	0.1	0.0	0.0	0.1	0.1	0.0	0.1	0.2	0.3
Foxtail millet	0.1	0.1	0.7	0.7	0.1	0.1	0.7	0.7	0.1	0.1	0.6	0.6	0.2	0.2	1.1	1.2
Glutinous rice	0.0	0.1	0.0	0.4	0.0	0.1	0.0	0.3	0.0	0.0	0.0	0.3	0.0	0.2	0.0	1.0
Gochujang	0.0	0.1	0.2	0.7	0.0	0.1	0.2	0.5	0.1	0.1	0.3	0.5	0.0	0.0	0.1	0.2
Job’s tears	0.0	0.0	0.1	0.1	0.0	0.0	0.1	0.1	0.0	0.0	0.1	0.1	0.0	0.0	0.0	0.0
Maize	0.2	0.3	1.0	1.2	0.2	0.2	0.9	1.1	0.2	0.3	1.1	1.2	0.5	0.5	2.5	3.0
Mixed paste	0.0	0.0	0.0	0.2	0.0	0.0	0.0	0.1	0.0	0.0	0.0	0.2	0.0	0.0	0.0	0.0
Mungbean	0.0	0.0	0.0	0.0	0.0	0.0	0.0	0.0	0.0	0.0	0.0	0.0	0.0	0.0	0.0	0.0
Noodle	0.1	0.4	0.6	2.4	0.1	0.3	0.6	1.8	0.1	0.3	0.7	2.0	0.1	0.2	0.5	1.6
Oat	0.0	0.0	0.0	0.0	0.0	0.0	0.0	0.0	0.0	0.0	0.0	0.0	0.0	0.0	0.0	0.0
Pea	0.0	0.0	0.0	0.0	0.0	0.0	0.0	0.0	0.0	0.0	0.0	0.0	0.0	0.0	0.0	0.0
Proso millet	0.0	0.0	0.0	0.0	0.0	0.0	0.0	0.0	0.0	0.0	0.0	0.0	0.0	0.0	0.0	0.0
Ramen	0.2	0.4	1.3	2.8	0.2	0.3	1.3	2.4	0.2	0.4	1.5	2.5	0.0	0.1	0.1	0.2
Red bean	0.0	0.0	0.0	0.0	0.0	0.0	0.0	0.0	0.0	0.0	0.0	0.0	0.0	0.0	0.0	0.0
Rice wine	0.0	0.1	0.0	0.4	0.0	0.1	0.0	0.3	0.0	0.1	0.0	0.3	0.0	0.0	0.0	0.0
Snack	0.0	0.1	0.1	0.8	0.0	0.1	0.1	0.6	0.0	0.1	0.1	0.5	0.1	0.5	0.3	2.4
Sorghum	0.1	0.1	0.4	0.4	0.1	0.1	0.4	0.4	0.1	0.1	0.3	0.3	0.1	0.1	0.7	0.7
Soy sauce	0.0	0.1	0.0	0.2	0.0	0.0	0.0	0.1	0.0	0.0	0.0	0.1	0.0	0.1	0.0	0.2
Soybean	0.0	0.0	0.0	0.2	0.0	0.0	0.0	0.2	0.0	0.0	0.0	0.1	0.0	0.0	0.0	0.2
Soybean paste	0.0	0.1	0.1	0.4	0.0	0.1	0.1	0.3	0.0	0.1	0.1	0.3	0.0	0.1	0.1	0.4
Soymilk	0.0	0.1	0.0	0.0	0.0	0.1	0.0	0.0	0.0	0.0	0.0	0.0	0.0	0.3	0.0	0.8
Tofu	0.0	0.3	0.0	1.6	0.0	0.2	0.0	1.2	0.0	0.2	0.0	1.1	0.0	0.6	0.0	3.0
Wheat	0.0	0.0	0.0	0.0	0.0	0.0	0.0	0.0	0.0	0.0	0.0	0.0	0.0	0.0	0.0	0.0
Wheat flour	0.1	0.2	0.4	0.9	0.1	0.1	0.4	0.8	0.1	0.1	0.5	0.8	0.1	0.1	0.3	0.7
White rice	0.0	2.4	0.0	7.3	0.0	1.7	0.0	5.3	0.0	1.5	0.0	4.4	0.0	4.3	0.0	11.0
Total	2.3	7.2	12.2	31.6	1.9	5.3	10.2	23.8	2.1	5.1	12.4	25.0	1.5	8.7	8.0	32.5

^1^ Average food intake; ^2^ 95% extreme food intake; ^3^ LB, minimum lower bound (<LOD = 0); ^4^ UB, maximum upper bound (<LOD = LOD); ^5^ Cereal-based baby formula; ^6^ Milk-based baby formula. TDI, tolerable daily intake; DON, deoxynivalenol; LOD, limit of detection.

**Table 2 toxins-15-00460-t002:** Estimated TDI% of NIV group by three scenarios according to glucoside bioavailability.

Food	Scenario 1				Scenario 2				Scenario 3 (Adult)				Scenario 3 (Infant)			
	Mean ^1^		95% ^2^		Mean		95%		Mean		95%		Mean		95%	
	LB ^3^	UB ^4^	LB	UB	LB	UB	LB	UB	LB	UB	LB	UB	LB	UB	LB	UB
Baby formula ^5^	0.0	0.0	0.0	0.0	0.0	0.0	0.0	0.0	0.0	0.0	0.0	0.0	0.4	1.3	2.3	7.8
Baby formula ^6^	0.0	0.0	0.0	0.0	0.0	0.0	0.0	0.0	0.0	0.0	0.0	0.0	0.0	0.0	0.0	0.0
Barley	1.1	1.3	5.5	6.6	1.0	1.1	5.0	5.7	0.9	1.0	4.7	5.2	1.3	1.5	6.3	7.4
Barley tea	0.0	0.1	0.0	0.4	0.0	0.1	0.0	0.3	0.0	0.1	0.0	0.3	0.0	0.1	0.0	0.2
Beer	0.0	1.1	0.0	5.7	0.0	0.9	0.0	4.5	0.0	1.1	0.0	7.5	0.0	0.0	0.0	0.0
Breakfast cereal	0.0	0.1	0.0	0.2	0.0	0.0	0.0	0.1	0.0	0.0	0.0	0.1	0.0	0.2	0.0	0.6
Brown rice	0.1	0.5	0.4	2.3	0.1	0.4	0.4	1.8	0.1	0.3	0.4	1.6	0.1	0.5	0.5	2.6
Buckwheat	0.0	0.0	0.0	0.0	0.0	0.0	0.0	0.0	0.0	0.0	0.0	0.0	0.0	0.0	0.0	0.0
Cheonggukjang	0.0	0.0	0.0	0.1	0.0	0.0	0.0	0.1	0.0	0.0	0.0	0.1	0.0	0.0	0.0	0.0
Chunjang	0.0	0.1	0.1	0.3	0.0	0.1	0.1	0.2	0.0	0.1	0.1	0.2	0.1	0.1	0.3	0.7
Foxtail millet	0.2	0.3	1.2	1.3	0.2	0.2	1.1	1.1	0.2	0.2	0.9	1.0	0.3	0.4	1.9	2.0
Glutinous rice	0.0	0.3	0.0	1.3	0.0	0.2	0.0	0.9	0.0	0.1	0.0	0.7	0.0	0.7	0.0	2.8
Gochujang	0.7	0.9	3.5	4.3	0.6	0.7	2.8	3.3	0.6	0.7	3.0	3.5	0.2	0.2	1.0	1.3
Job’s tears	0.1	0.1	0.2	0.2	0.1	0.1	0.2	0.2	0.1	0.1	0.1	0.1	0.0	0.0	0.0	0.0
Maize	0.1	0.2	0.3	0.9	0.1	0.2	0.3	0.7	0.1	0.2	0.3	0.7	0.1	0.4	0.7	2.1
Mixed paste	0.1	0.2	0.8	1.1	0.1	0.2	0.7	1.0	0.2	0.2	1.0	1.3	0.0	0.0	0.0	0.0
Mungbean	0.0	0.0	0.0	0.0	0.0	0.0	0.0	0.0	0.0	0.0	0.0	0.0	0.0	0.0	0.0	0.0
Noodle	0.0	0.8	0.0	5.1	0.0	0.6	0.0	3.6	0.0	0.5	0.0	3.5	0.0	0.4	0.0	3.3
Oat	0.0	0.1	0.0	0.1	0.0	0.0	0.0	0.1	0.0	0.0	0.0	0.1	0.0	0.0	0.0	0.1
Pea	0.0	0.0	0.0	0.1	0.0	0.0	0.0	0.0	0.0	0.0	0.0	0.0	0.0	0.0	0.0	0.0
Proso millet	0.0	0.0	0.0	0.0	0.0	0.0	0.0	0.0	0.0	0.0	0.0	0.0	0.0	0.0	0.0	0.1
Ramen	0.0	0.7	0.0	4.5	0.0	0.5	0.0	3.2	0.0	0.4	0.0	3.0	0.0	0.1	0.0	0.3
Red bean	0.0	0.0	0.0	0.1	0.0	0.0	0.0	0.1	0.0	0.0	0.0	0.0	0.0	0.0	0.0	0.1
Rice wine	0.0	0.2	0.0	0.8	0.0	0.2	0.0	0.6	0.0	0.2	0.0	0.7	0.0	0.0	0.0	0.0
Snack	0.0	0.4	0.2	2.3	0.0	0.3	0.2	1.7	0.0	0.2	0.2	1.4	0.2	1.4	0.8	6.8
Sorghum	0.1	0.1	0.4	0.5	0.1	0.1	0.4	0.4	0.1	0.1	0.3	0.3	0.1	0.1	0.7	0.8
Soy sauce	0.0	0.1	0.2	0.6	0.0	0.1	0.2	0.5	0.0	0.1	0.2	0.5	0.0	0.2	0.2	0.6
Soybean	0.0	0.1	0.1	0.7	0.0	0.1	0.1	0.5	0.0	0.1	0.1	0.5	0.0	0.1	0.1	0.5
Soybean paste	0.0	0.2	0.0	1.1	0.0	0.1	0.0	0.8	0.0	0.1	0.0	0.7	0.0	0.2	0.0	1.0
Soymilk	0.0	0.2	0.0	0.0	0.0	0.2	0.0	0.0	0.0	0.2	0.0	0.0	0.1	0.7	0.4	2.1
Tofu	0.0	0.9	0.0	4.7	0.0	0.6	0.0	3.3	0.0	0.6	0.0	3.0	0.0	1.7	0.0	8.5
Wheat	0.0	0.0	0.0	0.0	0.0	0.0	0.0	0.0	0.0	0.0	0.0	0.0	0.0	0.0	0.0	0.0
Wheat flour	2.7	2.9	15.5	16.7	2.1	2.2	12.3	13.0	2.1	2.3	12.5	13.3	2.2	2.4	11.5	12.3
White rice	0.0	6.9	0.0	20.8	0.0	4.9	0.0	14.8	0.0	4.1	0.0	12.4	0.0	12.3	0.0	31.1
Total	5.4	18.8	28.4	82.9	4.5	14.1	23.7	62.6	4.5	13.2	24.0	61.6	5.3	25.2	26.9	95.2

^1^ Average food intake; ^2^ 95% extreme food intake; ^3^ LB, minimum lower bound (<LOD = 0); ^4^ UB, maximum upper bound (<LOD = LOD); ^5^ Cereal-based baby formula; ^6^ Milk-based baby formula. TDI, tolerable daily intake; DON, deoxynivalenol; LOD, limit of detection.

**Table 3 toxins-15-00460-t003:** Relative bioavailability of glucoside conjugates.

No.	Species	Samples (*n*)	Route	Toxin	Bioavailability (%)	Relative Bioavailability (%)	Reference
1	Human (18–62 years)	20	Oral	DON,	64.0 ± 22.8	~100	[14]
				DON3G	58.2 ± 16.0		
2	Pig (11 weeks)	6	Oral	DON,	81.3 ± 17.4	~20	[15]
				DON3G	16.1 ± 5.4		
3	Pig (4 weeks)	8	Oral	DON,	77.9 ± 47.7	~100	[13]
				DON3G	83.0 ± 69.1		
4	Pig (4 weeks)	4	Oral	DON,	84.8 ± 9.7	~50	[41]
				DON3G	40.3 ± 10.0		
5	Rat (5 months)	6	Oral	DON,	14.9 ± 5.0	~25	[42]
				DON3G	3.7 ± 0.7		
6	Chicken (3 weeks)	6	Oral	DON,	5.6 ± 2.1	~70	[15]
				DON3G	3.8 ± 2.7		
7	Rat (6 weeks)	6	Oral	NIV,	3.7 ± 0.7	~35	[43]
				NIV3G	1.3 ± 0.3		

DON, deoxynivalenol; DON3G, deoxynivalenol-3-β-D-glucoside; NIV, nivalenol; NIV3G, nivalenol-3-β-D-glucoside.

## Data Availability

Not applicable.

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
