# Peer review of "Risk Assessment Considering the Bioavailability of 3-β-d-Glucosides of Deoxynivalenol and Nivalenol through Food Intake in Korea"

_toxins, 2023, doi:10.3390/toxins15070460_

Round 1

Reviewer 1 Report

Thi study conducted a risk assessment considering the bioavailability  of type B trichothecene glucoside conjugates, and found that DON and NIV exposure  was at a safe level, and infants were more susceptible to glucoside conjugate exposure than adults. 

Is is better to use statistics analysis to make the data more convincing?

Author Response

We appreciate the time and efforts by the editor and referees in reviewing this manuscript. We have addressed all issues indicated in the review report, and hope that the revised version can meet the journal publication requirements.

We send you the modify manuscript with all the changes marked in RED. Our responds to the comments are as follows:

Comment: This study conducted a risk assessment considering the bioavailability of type B trichothecene glucoside conjugates, and found that DON and NIV exposure was at a safe level, and infants were more susceptible to glucoside conjugate exposure than adults.

Is better to use statistics analysis to make the data more convincing?

Response: Thank you for your review on this study. First of all, we agree to utilize statistical analysis to increase the reliability of our data.

Exposure calculations based on a deterministic approach typically use the formula: (occurrence (μg/kg) × food intake (g)) / body weight (kg). The value of each parameter used at this time uses specific point values such as average occurrence, average food intake, and average body weight. For this reason, deterministic methods are also referred to as point estimation. As a result, exposures calculated using the deterministic method only contain information about specific values and no information about the distribution of exposure values is obtained. Therefore, statistical analysis of exposures calculated by deterministic methods is not applicable.

In the probabilistic method, after determining an appropriate distribution model using the occurrence, food intakes, and body weight of sample group, random values are repeatedly calculated using the distribution model. In this study, a database calculated through 100,000 iterations was used for statistical analysis. Student's t-test was used to compare the adult and infant groups, and the results are described in lines 253-259. The added descriptions read “In the case of deterministic risk assessment, statistical analysis could not be performed because there was no variance in the data due to the point estimation method. In the case of the probabilistic method, the TDI% of DON group through total food intake of the adult and infant were 16.2% ± 152.2% and 24.8% ± 153.1%, respectively, showing statistically significant differences (p<0.05). The TDI% of NIV group also showed a significant difference (p<0.05) between the adult (42.2% ± 379.1%) and infant (65.0% ± 356.3%)”. Accordingly, the method for statistical analysis was added in the Materials and Methods section (lines 392-596).

Author Response

We appreciate the time and efforts by the editor and referees in reviewing this manuscript. We have addressed all issues indicated in the review report, and hope that the revised version can meet the journal publication requirements.

We send you the modify manuscript with all the changes marked in RED. Our responds to the comments are as follows:

Comment 1: The present study provides a risk characterization derived from the dietary exposure to deoxynivalenol, nivalenol and their glucosides forms in Korean population. A combination between consumption data reported in the Seventh Korea National Health and Nutrition Examination Survey (KNHANES) and contamination data reported in a previous publication was performed for a deterministic approach, whereas a probabilistic approach was also conducted through a Monte Carlo simulation.

I think that the present investigation is not suitable for publication in Toxins due to severe flaws concerning, especially, the novelty. Authors got a dataset generated in previous publication by Lee et al. (2020), where another risk characterization is performed using the Fifth Korean National Health and Nutrition Examination Survey with similar results, so authors just updated the consumption data and added a probabilistic analysis that, as well indicated in the manuscript, presented problems when dealing with the high rate of negative samples. Besides, authors stated that the key contribution of this paper was considering the bioavailability of NIV3G and DON3G as a part of the risk characterization calculations. Three different scenarios were established according to the relative bioavailability reported for different species, except for NIV3G, whose bioavailability has only been assessed in rats. Altogether, I think the core of the present investigation is too weak.

Response 1: Although glucoside conjugates are less toxic by themselves, they can be hydrolyzed and converted to the parent toxin during human digestion (equivalent toxic potential to the parent toxin). For this reason, EFSA proposes to manage glucoside conjugates as group-TDI together with the toxin.

However, a clear absorption rate (bioavailability) for glucoside conjugate has not yet been confirmed. EFSA also proposes group-TDI on the assumption that 100% is hydrolyzed and absorbed. Therefore, the absorption rate of glucoside conjugate is considered to be a very important point in characterizing the risk of glucoside conjugate, a modified mycotoxin.

Risk assessment considering absorption rate has not been reported yet, and the risk assessment considering absorption rate attempted in this study is thought to be a novel idea.

Nevertheless, we are deeply grateful for your suggestions for improving the quality of this study. We faithfully reviewed and reflected your suggestion and responded below.

Comment 2: However, I appreciate the effort that authors made to perform and write the present manuscript, so I would like to make some comments about it:

  • The goal of the investigation is too vaguely described at the end of the introduction section. I suggest authors give some more brief details, such as the typology and number of samples, analytical/statistical techniques… This would help set, in a clearer way, the magnitude and reach of your manuscript.

Response 2: Thank you for your comment. In accordance with your comments, we have provided detailed descriptions of the data and methodology used to further clarify the scope of this study in the introduction. The corrections are described in lines 110-119. The corrected sentences read “In this study, three exposure scenarios were set, considering the absorption rate of glucoside conjugates in order to evaluate the risk of type B trichothecenes including glucoside conjugates. The risk was characterized using occurrence data of 537 cereals, legumes, and their processed products investigated in previous studies. For systematic risk assessment, exposure was estimated for each type of food processing, and both deterministic and probabilistic approaches (Monte Carlo simulation method) were used”. 

Comment 3: Even if the formal analysis is well explained and conducted, results are poorly discussed. Authors could compare the exposure of Korean population to other exposure assessment studies conducted in other countries with similar or different dietary patterns or even relate the outcomes to biomonitoring studies.

Response 3: To complement the lack of discussion in this study, as you suggested, we have added risk assessment results and comparisons in other regions with eating habits similar to Korea. Additions were made to lines 125-131. The added sentences read “In the case of the DON group, the exposure of 0.02–0.32 μg/kg bw/day calculated in this study was similar to or lower than 0.2–3.7 μg/kg bw/day evaluated by EFSA and 0.16–0.74 μg/kg bw/day evaluated by Gab-Allah et al in Egypt [2,20]. It was also similar to 0.39 μg/kg bw/day based on human urine biomonitoring data evaluated in Portugal [21]. In the case of the NIV group, the exposure of 0.02-0.33 calculated in this study was similar to the 0.05-0.23 μg/kg bw/day evaluated by Gab-Allah et al. for grain-based food in Egypt [20]”. Accordingly, two references (Ref 20 and 21) were newly added in the revised manuscript (lines 470 – 476).

In addition, the high exposure of infants, which was identified as one of the main results, was compared with the adult and statistical analysis was performed to increase the reliability of the results. The analysis results were added in lines 253-259. The added sentences read “In the case of deterministic risk assessment, statistical analysis could not be performed because there was no variance in the data due to the point estimation method. In the case of the probabilistic method, the TDI% of DON group through total food intake of the adult and infant were 16.2% ± 152.2% and 24.8% ± 153.1%, respectively, showing statistically significant differences (p<0.05). The TDI% of NIV group also showed a significant difference (p<0.05) between the adult (42.2% ± 379.1%) and infant (65.0% ± 356.3%)”. Accordingly, the method for statistical analysis was newly added in the Materials and Methods section (lines 390-596).

Reviewer 3 Report

The paper, titled “Risk Assessment Considering the Bioavailability of 3-5-D-Glucosides of Deoxynivalenol and Nivalenol through Food Intake in Korea”, is the first study to conduct a risk assessment considering the bioavailability of type B trichothecene glucoside conjugates. The topic can definitely attract the interest of the Toxins readership and the work brings elements of novelty. The paper is well-written, well-structured, clear, and concise. A few minor issues are listed below, however, this does not affect the publication of the manuscript in this journal.

1. Lines 46-55: I suggest indicating the location of the cited survey data, as the prevalence of mycotoxins is greatly influenced by geographical factors.

2. Fig 2: Please indicate the meanings represented by different colors on the bar chart.

Author Response

We appreciate the time and efforts by the editor and referees in reviewing this manuscript. We have addressed all issues indicated in the review report, and hope that the revised version can meet the journal publication requirements.

We send you the modify manuscript with all the changes marked in RED. Our responds to the comments are as follows:

The paper, titled “Risk Assessment Considering the Bioavailability of 3-5-D-Glucosides of Deoxynivalenol and Nivalenol through Food Intake in Korea”, is the first study to conduct a risk assessment considering the bioavailability of type B trichothecene glucoside conjugates. The topic can definitely attract the interest of the Toxins readership and the work brings elements of novelty. The paper is well-written, well-structured, clear, and concise. A few minor issues are listed below, however, this does not affect the publication of the manuscript in this journal.

Comment 1: Lines 46-55: I suggest indicating the location of the cited survey data, as the prevalence of mycotoxins is greatly influenced by geographical factors.

Response 1: In response to reviewer’s indication, we have added information about the region where the analyte sample was collected in the revised manuscript (lines 49-56).

Comment 2: Fig 2: Please indicate the meanings represented by different colors on the bar chart.

Response 2: Sorry for our mistake. Fig. 2 was missing the description of each color and pattern. Therefore, it has been replaced with a modified picture.

Reviewer 4 Report

The manuscript “Risk Assessment Considering the Bioavailability…” is well written, and it can be published in IJMS. However, there are some points to be stresses and/or clarified. The most important one can be summarized in the 

Abstract, lines 15-16: “The infants appeared to be most susceptible…” Your data presented here in the paper seem not clear to a nutritionist and food chemist as I am. And already in the abstract, his statement has to be supported with a statement that supports this fundamental fact. Additionally, the last sentence in the abstract (lines 18-19) is not clear.  

Similar objection – Conclusions, lines 276-277, and partially until the line 279 (it is clear, that the TDI% value shall be highest in the starting material). Where are concrete data that support it? Admittedly, I may not understand you, but I still do need more facts that address and support this important point. 

Further corrections.

Lines 37-39 – please add Reference(s)

Line 269 – please delete “first”. 

Lines 284-286 – the last sentence is not clear, or at least it is clumsy. 

Author Response

We appreciate the time and efforts by the editor and referees in reviewing this manuscript. We have addressed all issues indicated in the review report, and hope that the revised version can meet the journal publication requirements.

We send you the modify manuscript with all the changes marked in RED. Our responds to the comments are as follows:

Comment 1: The manuscript “Risk Assessment Considering the Bioavailability…” is well written, and it can be published in Toxins. However, there are some points to be stresses and/or clarified. The most important one can be summarized in the Abstract, lines 15-16: “The infants appeared to be most susceptible…” Your data presented here in the paper seem not clear to a nutritionist and food chemist as I am. And already in the abstract, his statement has to be supported with a statement that supports this fundamental fact.

Response 1: Thank you for your comments. We reviewed the abstract and recognized that its presentation may not be clear. That sentence has been modified in lines 15-17. The modified sentence reads “Notably, infants showed higher TDI% than adults for both toxin groups”.

Comment 2: Additionally, the last sentence in the abstract (lines 18-19) is not clear.  

Response 2: Regarding the last sentence, we thought that the explanation to consider the absorption rate (bioavailability) of glucoside conjugates is insufficient, so we added that glucoside conjugates are converted into parent toxins during digestion and absorption (lines 18-21). The added sentence reads “Since glucoside conjugates can be converted into parent toxins during the digestion process, risk assessment considering bioavailability allows evaluating the risk level of glucoside conjugates more accurately and can direct their safety management in the future”.

Comment 3: Similar objection – Conclusions, lines 276-277, and partially until the line 279 (it is clear, that the TDI% value shall be highest in the starting material). Where are concrete data that support it? Admittedly, I may not understand you, but I still do need more facts that address and support this important point.

Response 3: With the same content as the previous comment, the expression that infants are more vulnerable has been modified here as well (lines 296-298). The modified sentence reads “Since infants showed higher TDI% than adults for both toxin groups, a more caution is required”.

TDI% was calculated to characterize the risk according to the processing type. Calculation of the TDI% was expressed by comparing the exposure of each toxin group to the TDI, and two approaches were applied, based on deterministic methods (Figure 3) and stochastic methods (Figure 4).

In the deterministic method, in some cases, simple processed (S) and fermentation (F) groups were similar or higher than the raw material (R) group (Figure 3, B, 95%-LB scenario), but the overall trend was in the order of raw material (R) > simple processed (S) > fermentation (F).

In the probabilistic method, except for scenario 3 (adult) of the DON and NIV groups, the calculated TDI% was raw material > simple processed > fermentation (lines 232-242). In the case of scenario 3 (adult), TDI% of both toxin groups for simple processed (2.8% for the DON group and 7.0% for the NIV group) were slightly lower than those for fermentation (3.2% for the DON group and 9.2% for the NIV group). Thus, the overall trend can be described in the order of raw material > simple processed > fermentation.

Nevertheless, all trends were not constant, so "decreased" was modified to "tended to decrease" according to the reviewer's comment (lines 298-299).

Comment 4: Lines 37-39 – please add Reference(s) 

Response 4: In response to reviewer’s indication, two reference were added in the revised manuscript (lines 38-40).

Comment 5: Line 269 – please delete “first”.

Response 5: Thank you for your comment. The word "first" was deleted in the revised manuscript.

Comment 6: Lines 284-286 – the last sentence is not clear, or at least it is clumsy. 

Response 6: To help the reader's understanding, the explanation of the relationship between glucoside conjugate absorption rate and risk assessment has been modified (lines 289-291). The modified sentence reads “In the future, if accurate bioavailability is confirmed through accumulation of in vivo studies on absorption of glucoside conjugates, more accurate risk estimation will be possible”.

Round 2

Author Response

We really appreciate the time and efforts by the reviewer in reviewing this manuscript. Our responds to the comments are as follows:

Comment:

I truly appreciate the effort taken to improve the quality of the manuscript following the reviewers’ comments. You accurately addressed some of the points that I raised during the first round and I think this second version is a correct and well-written scientific manuscript. Nevertheless, I still think that the manuscript is not suitable for publication in Toxins (IF: 5.075; Q1 in Toxicology and Q2 in Food Science & Technology) not because of the quality of the manuscript but the research itself and its lack of scientific relevance.

As I previously told authors, I firmly believe that the results presented in the manuscript lack of scientific relevance for several reasons that I want to clarify.

First, the dataset has been borrowed from a previously published paper, as authors indicate in Line 343. In that paper, Lee et al. also performed an exposure assessment with that dataset but using the Fifth Korean National Health and Nutrition Examination Survey instead of the Seventh, that is the survey used in the present manuscript. As a result, the outcomes are similar between both studies, as expected. Exposure assessment studies that are based on total diet studies in combination with consumption data consider, by default, a total bioavailability of 100%, which provides worst-case scenario results.

As a novelty compared to Lee et al., authors also conducted a probabilistic approach using Monte Carlo simulation. As properly addressed by authors, this approach fails to provide an accurate estimation when dealing with a high percentage of left-censored data, which is a frequent fact when working with mycotoxins. Therefore, authors could only provide an estimation for the upper bound data.

The key contribution of the present manuscript was stated as:

 “This was the first study to conduct a risk assessment considering the bioavailability of type B trichothecene glucoside conjugates.”

In the conclusion section, authors discussed the limitations of this paper as follows:

“Nevertheless, a limitation of this risk assessment is that it is based on insufficient previous studies to definitively determine the bioavailability of glucoside conjugates.”

This means that the core of the research is sustained in weak assumptions: evaluating different bioavailability rates even if there is still a gap of knowledge regarding that matter. Out of all the bioavailability rates assayed in this manuscript, only one is extracted from human data. That rate is estimated in 100% for DON3G, which leads again to the way traditional exposure assessment studies are conducted. Consequently, the obtained results are similar to those previously reported by Lee et al.

Including bioavailability rates in exposure assessment studies will allow us to provide more accurate data about the impact of mycotoxins in human health, but first we must have a strong foundation to build upon. Therefore, I think using uncertain bioavailability rates calculations is not enough to justify a whole publication in a journal like Toxins. I want to strongly emphasize that my decision does not have anything to do with the manuscript, that I think is a correct scientific manuscript, but with the scientific relevance of the investigation.

Response:

Thank you very much for your review of our study.

As a limitation of this study, you commented on the lack of differentiation from the contents of the previous study (Lee et al.). You also mentioned that the lack of differentiation (novelty) comes from the following reasons: 1) In this study, as a result, the scenario assuming 100% hydrolysis and absorption is not different from previous studies, 2) The basis for absorption data of glucoside conjugates is insufficient.

First, the risk assessment conducted in the previous study (Lee et al.) was based on information reported by EFSA in 2017. Since there was no experimental evidence for hydrolysis and absorption of glucoside conjugates in humans, EFSA (2007) conservatively assumed 100% absorption (worst case). Since then, the results of human studies by Vidal et al. have been reported.

When conducting a risk assessment, the clearer the data on the target substance, the more accurate the risk level can be estimated. However, there are not many cases in which data on the toxicity, etc., of target substances are perfectly prepared. Since most risk assessments are conducted under many assumptions and uncertainties based on research results accumulated at the time of risk assessment, it is appropriate to clearly describe them. Even if the same results were obtained assuming 100% absorption in the risk assessment conducted in the previous study and this study, there is a difference in terms of reliability between the one that was simply conservatively set at 100% and the one that was set based on human data (Even if there is only one study, it meets quality of the study). Similarly, EFSA or JECFA has been reevaluating health-based guidance values or risks by reflecting new toxicological data on the same substance.

In addition, in this study, in order to compensate the uncertainty caused by the lack of data, the collected data was classified according to priority into 1) human data, 2) animal data, and 3) in vitro data, and the absorption rate was reviewed. Then, based on human data, which has the highest priority, we set scenario 1 assuming 100% absorption. According to the priority, Scenario 2 assuming 50% absorption was established based on the data of pigs, which are not human data but have similar susceptibility to humans. And Scenario 3 (adults 25%, infants 75%) was established considering the difference in absorption rate according to age identified in the pig data.

Considering the human data (100% absorption), the risk assessment results in this study, of course, resulted in similar results to the classically performed case of 100% absorption. However, when considering the difference in absorption rate according to age (adults 25%, infants 75%) and estimating the risk considering intake and weight, the exposure of the infant group was higher than that of the adult group. This means that absorption rates of glucoside conjugates may differ between adults and infants.

The study of Vidal et al. was only conducted on adults due to ethical issues, and it is considered very difficult to conduct research on infants in the future. Therefore, data from studies in models that better mimic humans are of great importance for assessing age-dependent glucoside conjugate risk.

Recent papers on risk assessment of modified mycotoxin emphasize the importance of bioavailability for scientific and systematic risk assessment (Lorenz et al., 2019). In particular, in infants, the importance of bioavailability is more emphasized because the immune system, such as intestinal immunity, is not fully mature, so toxic dynamics may be different from those of adults (Rebellato et al., 2021).

In this study, a risk assessment was performed considering the absorption rate by integrating studies conducted after the EFSA report in 2017. Unlike the previous study (Lee et al.), which simply assumed 100% absorption rate, the difference in absorption rate between adults and infants was considered to reflect the fact that the absorption rate was different between adults and infants in the pig study. The results (TDI%) showed higher risk to infants in both deterministic and probabilistic methods. This result can be the basis for the need for research on the absorption of glucoside conjugates and can increase the motivation of researchers for absorption research, which is currently lacking.

Taking all of the above into consideration, a sentence was modified to emphasize the importance of risk assessment considering bioavailability in various scenarios in the revised manuscript (Line 102). The modified sentence reads “To estimate the health risk arising from the consumption of glucoside conjugates of DON and NIV, risk assessment considering bioavailability in various scenarios, which are much more realistic, is necessary [16,17]”. Additionally, two references [16, 17] were added in the revised manuscript (Line 102). Reference numbers due to the addition of references were corrected throughout the manuscript.

I would like to thank for reviewer’s comments again. I do hope we have addressed all points to reviewer’s satisfaction.